# Novel Triazeneindole Antibiotics: Synthesis and Hit-to-Lead Optimization

**DOI:** 10.3390/ijms26051870

**Published:** 2025-02-21

**Authors:** Boris Sorokin, Alla Filimonova, Anna Emelianova, Vadim Kublitski, Artem Gvozd, Vladimir Shmygarev, Ilia Yampolsky, Elena Guglya, Evgeniy Gusev, Denis Kuzmin

**Affiliations:** 1Moscow Center for Advanced Studies, Kulakova Str. 20, 123592 Moscow, Russia; sorokin.ba@gmail.com (B.S.); denisk@list.ru (D.K.); 2Kurchatov Centre for Genome Research, National Research Centre “Kurchatov Institute”, 123182 Moscow, Russia; allafil@yandex.ru; 3Institute for Personalized Oncology, I.M. Sechenov First Moscow State Medical University, 119991 Moscow, Russia; 4Shemyakin and Ovchinnikov Institute of Bioorganic Chemistry, Russian Academy of Sciences, 117997 Moscow, Russia; vkublitski@gmail.com (V.K.); gwozd_art@mail.ru (A.G.); vishmygarev@gmail.com (V.S.); ivyamp@ibch.ru (I.Y.); eguglya@gmail.com (E.G.); 5Severtsov Institute of Ecology and Evolution, Russian Academy of Sciences, Leninsky Prospect 33, 119071 Moscow, Russia; algogus@yandex.ru

**Keywords:** MRSA, antibiotic, triazeneindole, hit-to-lead, ADME-Tox, ABSSSI

## Abstract

Bacterial antibiotic resistance represents a major healthcare problem. In 2019, 4.95 million deaths were associated with antibiotic resistance, and it is estimated that, by 2050, up to 3.8% of the global gross domestic product could be lost due to this problem. Methicillin-resistant *Staphylococcus aureus* is one of the leading sources of hospital-acquired infections associated with increased mortality, length of hospital stay, and higher cost of treatment. Here, we describe the de novo synthesis of a library of 22 triazeneindole derivatives with high activity against a wide panel of multidrug-resistant MRSA clinical isolates. Leading compound BX-SI043 (ethyl 6-fluoro-3-[pyrrolidin-1-yl-azo]-1H-indole-2-carboxylate) showed high activity (minimal inhibitory concentration range, 0.125–0.5 mg/L) against 41 multidrug-resistant MRSA strains, as well as relatively low in vitro cytotoxicity (selectivity index, 76) and in vivo acute toxicity (maximum tolerated dose, 600 mg/kg), via intragastric administration in rats. These data suggest that BX-SI043 is a promising drug candidate for the development a novel MRSA treatment.

## 1. Introduction

Methicillin-resistant *Staphylococcus aureus* (MRSA) strains caused over 100 thousand direct deaths in 2019, being the third most deadly antibiotic-resistant bacteria globally and the most deadly antibiotic-resistant bacteria in the high-income region [1,2]. MRSA is one of the leading sources of hospital-acquired infections, and is associated with increased mortality, length of hospital stay, and cost of treatment. Acute bacterial skin and skin structure infections (ABSSSIs) are defined as bacterial skin/skin structure lesions with an area of at least 75 cm^2^ [3]. They encompass a wide spectrum of clinical presentations, ranging from mild symptoms to complicated, life-threatening diseases [4]. While these rates are reported to have declining trends, the mean cost of ABSSSI-related emergency department visits in the US has more than doubled in adults and tripled in children over the last two decades [5]. ABSSSIs are important reasons for both ambulatory visits and hospital admissions and place a heavy burden on the healthcare system. Almost 10% of all hospital admissions in the USA and nearly 15% of all infections treated in Europe are because of ABSSSIs [6]. ABSSSIs are primarily monomicrobial infections caused by aerobic Gram-positive cocci, particularly *Staphylococcus aureus*, including MRSA [7,8]. In some areas, the prevalence of drug-resistant *S. aureus* strains reaches 40–70% [9,10,11]. Patients with MRSA frequently present with a previous history of MRSA infection, advanced age, chronic open wounds, underlying chronic disease, and repeated contact with a healthcare facility [2,3,11]. Compared with hospital-associated MRSA, community-acquired MRSA tends to be more virulent and may carry genes that encode various exotoxins that are associated with tissue necrosis and a greater severity of disease [12,13]. According to current guidelines from the Infectious Diseases Society of America, recommended antibiotics for MRSA infection treatment are vancomycin, linezolid, clindamycin, daptomycin, ceftaroline, and trimethoprim-sulfamethoxazole [14]. However, many of these drugs are losing their clinical effectiveness due to emerging bacterial resistance [15,16,17,18,19]. Altogether, this makes the development of new anti-MRSA drugs for treatment of ABSSSI a promising field of research.

Triazeneindoles are promising compounds for the development of antimicrobial drugs. They exhibit potent in vitro activity against *Mycobacterium tuberculosis* [20,21,22], while structurally-related indole and triazene compounds show antibacterial [23,24,25,26,27], antifungal [28], and anticancer [26] activities. The presence of two functional groups with distinct antimicrobial activities in triazeneindoles suggests a possible hybrid mechanism of action, which would help to combat bacterial resistance mechanisms. As demonstrated in the present work, the synthesis and modification of triazeneindoles is relatively easy and straightforward, allowing for the development of a wide panel of clinically relevant compounds with diverse pharmacological activities.

This paper describes the hit-to-lead optimization of triazeneindole compounds used in the search for novel antibiotics against MRSA. Parent compound BX-SI001 was previously isolated from the cultural liquid of *Dunaliella salina* microalgae contaminated by *Streptomyces* sp. filamentous fungi, and it showed moderate antimicrobial activity against Gram-positive bacteria, including MRSA. We employed de novo synthesis to create a library of 21 derivatives of BX-SI001 and studied their antibacterial activity and cytotoxicity in vitro to select a leading molecule for ADME and animal toxicology studies.

## 2. Results

### 2.1. Library Design

For the initial library design, we identified the following antibacterial drug-likeness parameters [29] which could be analyzed in silico: molecular weight < 500, logP_oct/wat_ < 5, topological polar surface area > 70, number of hydrogen-bond donors < 7, number of hydrogen-bond acceptors < 12, and number of rotatable bonds < 8. Then, we screened the literature and drug databases for functional scaffolds that could possibly enhance the antibacterial activity or pharmacokinetic properties of the triazeneindole core. Using this data, we created a preliminary virtual library of 54 structures (Appendix A). Considering drug-likeness parameters and synthesis complexity, we have chosen 22 compounds for the final library (Figure 1).

### 2.2. Library Synthesis

We developed a method of synthesis of a wide variety of triazeneindole derivatives using four precursors that can be transformed to final compounds through a relatively simple two-stage reaction. The chemical structures of the compounds were evaluated using ^1^H NMR and ^13^C NMR (Appendix A). The reaction routes for the synthesis of the precursors and tested compounds are provided in Figure 2, Figure 3, Figure 4 and Figure 5. The water solubility of each synthesized compound was tested using HPLC and widely ranged from 0.00001 to 0.9 g/L (Appendix A).

### 2.3. Antimicrobial Activity and Cytotoxicity of Tested Compounds

The antimicrobial activity of 22 synthesized compounds was evaluated against eight clinical isolates of MRSA with multiple antibiotic resistance (Appendix A) using the broth dilution method, which is widely accepted as a standard method of antimicrobial activity evaluation. Most of the derivatives showed similar or lower activity in comparison to that of the original BX-SI043 molecule, but seven compounds exhibited 2–4 times lower MIC values. In parallel with the study of activity, the in vitro toxicity on human fibroblast cell lines was also evaluated. The effect of the compounds on cell viability was investigated using the MTT assay, which allows for the estimation of the total activity of the mitochondrial respiratory enzymes. This method is widely used to study the toxic effects of various compounds on cells, including at the screening stage for drug prototype substances. The IC50 concentrations of most compounds were similar to those of the original molecule and ranged from 10 to 25 mg/L. Seven compounds were found to be non-toxic to HEF cells in the tested concentrations, but they also showed poor antibacterial activity. The results of the experiments are presented in Table 1. After evaluation of antimicrobial activity and cytotoxicity, we selected the six most potent compounds for the next stage of the experiment.

### 2.4. Selectivity Index of the Most Potent Compounds

The structures and antimicrobial activity of the most potent compounds are presented in Figure 6. All molecules contained a fluorine atom in the C-6 position of the indole ring, which suggests a key role of this modification in the increase in antimicrobial activity. BX-SI043 and BX-SI045 showed the lowest MIC values (0.25 mg/L) against most tested strains.

To address the potential hepatotoxicity, the IC50 of the original compound and the most potent derivatives was evaluated for HepG2 human liver carcinoma cells. The selectivity index was calculated for each compound as the average IC50 on eukaryotic cells divided by the average MIC on MRSA (Table 2).

Compound BX-SI043 demonstrated the highest selectivity index (76) and the most potent activity (0.25 mg/L against all tested strains). Consequently, its antimicrobial activity was further evaluated on a broader panel of 41 multidrug-resistant MRSA clinical isolates (Appendix A), where it exhibited MIC values that were 4–8 times lower compared to those of the original molecule (Appendix A). BX-SI043 was chosen as a leading candidate for further ADME and toxicology studies.

### 2.5. Stability of the Leading Compound in Artificial Stomach and Intestinal Juices

Oral administration of drugs is the most common route for drug delivery in modern medicine due to its convenience, non-invasiveness, and patient compliance. The bioavailability of such drugs primarily depends on their stability in the gastrointestinal tract and during the first pass through the liver. Experiments using artificial gastric and intestinal juices revealed that BX-SI043 is rapidly metabolized in acidic gastric juice (t_1/2_ < 10 min) and fully metabolized in intestinal juice after 4 h of incubation (t_1/2_ = 1 h) (Appendix A). HPLC-MS data suggests that in the acidic conditions of artificial gastric juice, the molecule was decomposed to a diazonium fragment and pyrrolidine (Figure 7).

### 2.6. Stability of Leading Compound in Liver Microsomes

To assess the rate of drug metabolism in the liver, we used subcellular fractions of the liver, so called microsomes, that contain the main metabolic enzymes, such as cytochromes P450, flavin monooxygenases, UDP-glucuronosyltransferases, carboxylesterases, hydrolases, and others. The availability of the microsomes obtained from the liver of various mammalian species makes them one of the widely used tools for assessing the metabolism of new chemical compounds. Experiments showed that BX-SI043 is rapidly metabolized in the oxidation phase, with a half-life of 2.73 min in human liver microsomes and 1.4 min in rat liver microsomes (Appendix A).

### 2.7. Inhibition of Cytochrome Activities by the Leading Compound

The evaluation of drug metabolism by some of the cytochrome P450 isoforms is crucial for the prognosis of potential adverse effects due to drug–drug interactions. Specific metabolic substrates for microsomal cytochromes are used to assess the compound’s effect on liver cytochromes functions. Experiments using human liver microsomal fractions and substrates for seven isoforms of the cytochrome p450 demonstrated that BX-SI043 and its metabolites do not inhibit cytochrome isoforms activity (Appendix A).

### 2.8. Cellular Barrier Penetration of the Leading Compound

We used Caco-2 cells plated on the permeable membrane to measure permeability through cellular barriers. The Caco-2 cells have been successfully used to assess permeability and absorption in the gastrointestinal tract, as well as to study the active transport of drug candidates [30]. Caco-2 epithelial adenocarcinoma cells are morphologically and functionally similar to the intestinal barrier epithelium and also express a number of transporters, including P-glycoprotein (Pgp). Data on Caco-2 cell permeability and Pgp binding are also used to predict drug–drug interactions. Apparently, the permeability of BX-SI043 is relatively low, lower than that of ranitidine and rhodamine with cyclosporin A, indicating that the transport is likely due to passive diffusion through the cell layer, not to active transport. Notably, the different mass balance at the apical (49) and basolateral (25.1) sites suggests different stabilities of the BX-SI043 at the different compartments of the system due to unknown mechanisms. This can probably account for the asymmetry index of 0.37 that partially reflects the higher degradation rates in the different chambers. The results are presented in Appendix A.

### 2.9. Stability of the Leading Compound in Rat and Human Plasma and Binding to Plasma Proteins

Blood and plasma stability play an important role in drug discovery and development. Unstable compounds tend to have rapid clearance and a short half-life, resulting in poor in vivo performance. Protein binding influences the bioavailability and distribution of active compounds and is a limiting factor in the passage of drugs across biological membranes and barriers [31]. Stability studies in human and rat plasma revealed that the BX-SI043 is metabolized in human plasma by about 50% after 1 h and by 97% after 4 h of incubation. The compound is even more unstable in rat plasma, with only 1.7% remaining after 15 min of incubation (Appendix A). The binding of BX-SI043 to plasma proteins could not be determined because the compound was not stable in the pooled human plasma for the duration of the experimental protocol.

### 2.10. Investigation of Potential Cardiotoxicity of the Leading Compound Due to hERG Blockade

Blocking the voltage-dependent K^+^ channel of the heart (hERG) by drugs might lead to impaired myocardial repolarization, increased QT interval duration, and arrhythmia [32,33,34]. Therefore, test systems have been developed to study the activity of this channel in the presence of drugs and small molecules [32,35]. BX-SI043 and its metabolites are characterized by a weak inhibition of the hERG-dependent potassium channels, with IC50 > 50 μM. (Appendix A). Thus, BX-SI043 does not inhibit hERG-dependent potassium channels and most likely, does not exhibit cardiotoxicity.

### 2.11. Single-Dose Acute Toxicity of the Leading Compound in Rats

Intragastric administration of drugs to rats is often used to model oral administration in humans because it achieves high correlation of results [36]. We performed in vivo single dose acute toxicity studies of the leading compound BX-SI043 on Wistar outbred rats via the intragastric administration route. The first group (Group 1) of animals was the control group, and Groups 2–5 received various doses of the tested substance. Initially, based on in vitro toxicity data, Group 2 received a 300 mg/kg dose. After 24 h, no deaths occurred, so the next group (Group 3) received the maximum allowed dose of 2000 mg/kg. The deaths of animals in Group 3 occurred 3 days after substance administration, so Group 4 and Group 5 received lower doses of 1000 mg/kg and 600 mg/kg, respectively, to establish the maximum tolerated dose. Signs of intoxication were recorded in the 2000 mg/kg and 1000 mg/kg groups starting from the 4th day after administration. They included behavioral inhibition and a change in reaction to stimuli. In the 2000 mg/kg group, ruffled fur and nasal discharge were observed. Starting from the 9th day of the experiment, the condition, behavior, and appearance of all alive animals that received the BX-SI043 did not differ from those of the control group. Substance administration led to a lower food intake in the 2000 mg/kg and 1000 mg/kg groups starting from the 2nd day, but did not affect the water intake of the animals. Deaths of animals were recorded between 3 and 8 days after administration in the 2000 mg/kg and 1000 mg/kg groups. A summary of the effects of BX-SI043 on the tested animals, based on the 14-day follow-up after administration, is presented in Table 3.

The masses of the animals before the beginning of the experiment did not differ between groups. On the 2nd, 7th, and 15th days of the experiment, one-way ANOVA revealed no influence of the factor “group” on the body weight of the male and female rats (*p* > 0.05). However, it should be noted that females and males who received the substance in doses of 1000 and 2000 mg/kg on the 7th and 15th days of the experiment, respectively, showed a tendency to decrease body weight compared to the results for the control group (Appendix A). The immediate cause of death for most animals was acute heart failure. Males died more often than females—4 and 2 animals, respectively. During necropsy, pathological changes were revealed in the stomach of animals that received the compound in doses of 600, 1000, and 2000 mg/kg. The primary data for the relative organ weight conformed to the normal distribution. One-way ANOVA revealed no statistically significant differences in relative organ weights on the 15th day of the experiment between the groups that received the control substance and the test substance intragastrically in all doses (*p* > 0.05, Appendix A). Thus, there was no effect of intragastric administration of the test substance on the relative organ weights of the animals. Histologically, in the dead animals which received the compound at doses of 1000 and 2000 mg/kg, pathological keratinization of the epithelium in the nonglandular stomach, with blood soaking of the affected areas; hyperemia of the mucous membrane; as well as an admixture of blood in the intestinal contents were noted. In the group that received the compound at a dose of 2000 mg/kg, erosive lesions of the gastric mucosa were detected in one case. The results obtained indicate the presence of a local irritant effect of the BX-SI043 on the stomach. Administration of the BX-SI043 in the dosages of 600 mg and 300 mg per 1 kg of body weight did not lead to the death of animals or changes in their behavior and body weight. Thus, the maximum tolerated dose during intragastric administration of the BX-SI043 in rats was established as 600 mg/kg.

## 3. Discussion

The development of novel antibiotics is an important task, given the spreading antibiotic resistance. Our study showed that triazeneindole derivatives can be a promising platform for developing potent antibiotics against Gram-positive bacteria, specifically MRSA. Other researchers have found that similar triazeneindoles display good in vitro activity against mycobacteria [20,21,22] and fungi [28], identifying them as possible wide-spectrum drugs against different infections. All synthetized triazeneindole derivatives showed good in vitro activity against a wide panel of multidrug-resistant MRSA strains, while the most active molecule, BX-SI043, revealed a 4–8 times lower MIC than either vancomycin or linezolid, and it was active against 41 different multidrug-resistant clinical isolates.

The biological activity of triazeneindoles can be modulated by the inclusion of different functional groups, and our synthesis method allows us to produce various derivatives using relatively simple reactions. All six compounds with the highest in vitro activity against MRSA contained a fluorine atom in the C-6 position of the indole ring, which makes Precursor 3 the best candidate for the synthesis of future libraries. We believe that this effect is linked to one of the proposed mechanisms of the antibacterial activity of the triazeneindole molecules.

One of the mechanisms of the antibacterial activity of indole derivatives is the inhibition of bacterial DNA gyrase [23,24,25]. The increase in the activity of the C-6 fluorinated triazeneindole derivatives observed in our experiments supports these findings because it was shown that C-6 fluorination greatly increases the binding strength of quinolones, DNA gyrase inhibitors [37,38,39] that exhibit a high structural similarity to indoles. However, taking into account that the tested compounds were active against fluoroquinolone resistant strains, DNA gyrase inhibition may be not the sole mechanism of action.

Triazene moiety has also shown antibacterial and anticancer activities due to the disruption of DNA synthesis by diazonium metabolites [26,27]. A recent publication [28] revealed that triazeneindoles also exhibit antifungal activity through the formation of diazonium species that inhibit fungal fatty acid synthase 1. Interestingly, our stability studies showed that in acidic conditions, BX-SI043 rapidly decomposes to a diazonium metabolite similar to the active fragment mentioned in that publication, so the molecule’s instability in physiological fluids may not cause the decline of its in vivo antimicrobial activity. Thus, we propose that triazeneindoles may exhibit a hybrid mechanism of antimicrobial action, based on both the triazene and indole moieties.

Leading candidate BX-SI043 showed a good in vitro and in vivo toxicology profile. It displayed a relatively high selectivity index of 76, did not inhibit liver cytochrome activities, and showed no signs of cardiotoxicity on the hERG model. The substance was relatively well tolerated by animals, causing visible signs of intoxication and deaths only at doses higher than 1000 mg/kg. Necropsy revealed some local irritant effects of the tested compound on the animal stomach at concentrations above 600 mg/kg, but as this concentration showed no other ill-effects on animals, it was considered as the maximum tolerated dose.

The next step in our research would be in vivo efficiency studies using a rat model, but we will likely change the route of administration. Despite relatively low oral toxicity, the pharmacokinetic properties of BX-SI043—low water solubility, low stability in gastric and intestinal juices, and low cell membrane permeability—make peroral administration sub-optimal without modifying the molecule. Moreover, studies have shown that systemic antibiotics might be less effective against MRSA-caused ABSSSI than are topical drugs [40].

We see several advantages of using a topical form for BX-SI043 development. Firstly, it excludes the prolonged exposition to rat plasma and the gastric and intestinal juices, in which the molecule is highly unstable. Next, there are relatively few topical anti-MRSA drugs—fusidic acid, mupirocin and retapamulin—that already face resistance pressure, so novel topical antibiotics could easily find their market niche. Lastly, given that similar triazeneindoles show potent antifungal activity, the development of BX-SI043 could produce a drug candidate for treatment of a wide spectrum of skin infections. Given these considerations, efficiency studies on a rat model of MRSA-caused ABSSSI with a topical route of administration appears to be an optimal developmental route for BX-SI043.

Given their potent in vitro activity and their relatively easy synthesis and modification routes, triazeneindoles could become a promising platform for the synthesis of various compounds with antibacterial, antifungal, and anticancer activities. The introduction of different functional groups to indole ring and/or triazene moiety could significantly alter the pharmacological activity and bioavailability of triazeneindoles in order to overcome current stability limitations and broaden the spectrum of activity. Thus, further research and structure–activity optimization could lead to the development of a diverse portfolio of clinically-relevant triazeneindoles.

## 4. Materials and Methods

### 4.1. Synthesis of Tested Compounds

#### 4.1.1. Synthesis of Precursors 1–4

Precursor 1: A total of 11.8 g (0.1 mol) of 2-aminobenzonitrile, 33.2 g (0.2 mol) of bromoacetic acid ethyl ester, 41.1 g (0.3 mol) K_2_CO_3_, and 250 mL of N-methyl-2-pyrrolidone were added to a 500 mL round-bottomed flask. The reaction mixture was stirred for 6–8 h at 100 °C. After cooling, it was treated with 1000 mL of water and 200 mL of ethyl acetate. The organic layer was separated, dried over anhydrous K_2_CO_3_, and evaporated to a volume of 50 mL. The residue was passed through a 5 × 10 cm layer of silica gel in the 1:2 ethyl acetate–hexane system to separate resinous impurities. Fractions containing ethyl N-(2-cyanophenyl)glycinate were evaporated, dissolved in 150 mL of absolute tetrahydrofuran, and treated with 11.2 g (0.1 mol) of potassium tert-butoxide. The reaction mixture was stirred at room temperature for 16 h, evaporated, and treated with 200 mL of a saturated solution of NaHCO_3_ in water and 200 mL of ethyl acetate. The organic layer was dried over anhydrous K_2_CO_3_ and evaporated. The residue was purified on a 5 × 20 cm layer of silica gel in the 1:2 ethyl acetate–hexane system. Fractions containing Precursor 1 (controlled by LC/MS) were evaporated, and the residue was crystallized from 20 mL of methylene chloride. The yield after two stages was 3.7 g (18%).

Precursor 2: A total of 13.8 g (0.1 mol) of 2-chloropyridine-3-carbonitrile, 30.7 g (0.2 mol) of N-methylglycine ethyl ester hydrochloride, 41.1 g (0.3 mol) K_2_CO_3_, and 250 mL of N-methyl-2-pyrrolidone were added to a 500 mL round-bottomed flask. The reaction mixture was stirred for 6–8 h at 100 °C. After cooling, it was treated with 1000 mL of water and 200 mL of ethyl acetate. The organic layer was separated, dried over anhydrous K_2_CO_3_, and evaporated to a volume of 50 mL. The residue was passed through a 5 × 10 cm layer of silica gel in the 1:2 ethyl acetate–hexane system to separate resinous impurities. Fractions containing ethyl N-(3-cyano-2-pyridinyl)-N-methylglycinate were evaporated, dissolved in 150 mL of absolute tetrahydrofuran, and treated with 11.2 g (0.1 mol) of potassium tert-butoxide. The reaction mixture was stirred at room temperature for 16 h, evaporated, and treated with 200 mL of a saturated solution of NaHCO_3_ in water and 200 mL of ethyl acetate. The organic layer was dried over anhydrous K_2_CO_3_ and evaporated. The residue was purified on a 5 × 20 cm layer of silica gel in the 1:2 ethyl acetate–hexane system. Fractions containing Precursor 2 (controlled by LC/MS) were evaporated, and the residue was crystallized from 20 mL of methylene chloride. The yield after two stages was 4.6 g (21%).

Precursor 3: A total of 13.9 g (0.1 mol) of 2,4-difluorobenzonitrile, 27.8 g (0.2 mol) of glycine ethyl ester hydrochloride, 41.1 g (0.3 mol) K_2_CO_3_, and 250 mL of N-methyl-2-pyrrolidone were added to a 500 mL round-bottomed flask. The reaction mixture was stirred for 6–8 h at 100 °C. After cooling, it was treated with 1000 mL of water and 200 mL of ethyl acetate. The organic layer was separated, dried over anhydrous K_2_CO_3_, and evaporated to a volume of 50 mL. The residue was passed through a 5 × 10 cm layer of silica gel in the 1:2 ethyl acetate–hexane system to separate resinous impurities. Fractions containing ethyl N-(2-cyano-5-fluoro-phenyl)glycinate were evaporated, dissolved in 150 mL of absolute tetrahydrofuran, and treated with 11.2 g (0.1 mol) of potassium tert-butoxide. The reaction mixture was stirred at room temperature for 16 h, evaporated, and treated with 200 mL of a saturated solution of NaHCO_3_ in water and 200 mL of ethyl acetate. The organic layer was dried over anhydrous K_2_CO_3_ and evaporated. The residue was purified on a 5 × 20 cm layer of silica gel in the 1:2 ethyl acetate–hexane system. Fractions containing Precursor 3 (controlled by LC/MS) were evaporated, and the residue was crystallized from 20 mL of methylene chloride. The yield after two stages was 1.5 g (7%).

Precursor 4: A total of 13.8 g (0.1 mol) of 2-chloropyridine-3-carbonitrile, 30.7 g (0.2 mol) of glycine ethyl ester hydrochloride, 41.1 g (0.3 mol) K_2_CO_3_, and 250 mL of N-methyl-2-pyrrolidone were added to a 500 mL round-bottomed flask. The reaction mixture was stirred for 6–8 h at 100 °C. After cooling, it was treated with 1000 mL of water and 200 mL of ethyl acetate. The organic layer was separated, dried over anhydrous K_2_CO_3_, and evaporated to a volume of 50 mL. The residue was passed through a 5 × 10 cm layer of silica gel in the 1:2 ethyl acetate–hexane system to separate resinous impurities. Fractions containing ethyl N-(3-cyano-2-pyridinyl)glycinate were evaporated, dissolved in 150 mL of absolute tetrahydrofuran, and treated with 11.2 g (0.1 mol) of potassium tert-butoxide. The reaction mixture was stirred at room temperature for 16 h, evaporated, and treated with 200 mL of a saturated solution of NaHCO_3_ in water and 200 mL of ethyl acetate. The organic layer was dried over anhydrous K_2_CO_3_ and evaporated. The residue was purified on a 5 × 20 cm layer of silica gel in the 1:2 ethyl acetate–hexane system. Fractions containing Precursor 4 (controlled by LC/MS) were evaporated, and the residue was crystallized from 20 mL of methylene chloride. The yield after two stages was 4.9 g (24%).

#### 4.1.2. Synthesis of Compounds BX-SI001, 003, 005, 010, 016, 055

A total of 0.22 g (0.001 mol) of Precursor 1, 5 mL of water, and 5 mL of dimethylformamide were placed in a round-bottom flask with a volume of 50 mL. The reaction mixture was cooled to 5 °C using an ice bath. Then, 0.3 mL of concentrated hydrochloric acid was added. After 15 min, the reaction mixture was treated with 0.086 g (0.00125 mol) of sodium nitrite and stirred at this temperature for 30 min. After that, 0.01 mol of the corresponding secondary amine 35–48 was added to the mixture, and it was heated at 50° C for 1 h. After cooling, the reaction mixture was diluted with 25 mL of water. The formed precipitate was filtered and dried. NMR spectra of compounds were acquired using a Bruker Fourier 300 NMR spectrometer (Bruker Biospin, Ettlingen, Germany) (1H, 300.17 MHz and 13C, 75.47 MHz) in d6-DMSO (Solvex, Saint-Petersburg, Russia) solutions in standard 5 mm tubes at 25 °C. Residual solvent peaks were used as references (2.50 ppm and 39.5 ppm for ^1^H and ^13^C NMR for d6-DMSO, respectively). High-resolution mass spectra of the synthetic compounds were obtained using a QExactive Plus Orbitrap mass-spectrometer (ThermoScientific, St. Louis, MO, USA). The samples were injected directly into the ESI ion source using a syringe into a solution of 0.1% FA/90% acetonitrile/10% H_2_O. The MS data were collected in DDA mode, with the spectra recorded in positive and negative modes, with a 150–500 mass range at 70 K resolution for MS1 and a 17.5 K resolution at 30 (N)CE, with a 1.4 *m*/*z* isolation window, for MS2. The melting point was obtained using a FP62 melting point apparatus (Mettler Toledo, Greifensee, Switzerland).

BX-SI001: Ethyl (E)-3-((4-methylpiperazin-1-yl)diazenyl)-1H-indole-2-carboxylate; yield 29%; mp: 138 °C (from DMF-H_2_O); ^1^H NMR (300 MHz, DMSO-*d*_6_) δ 11.65 (s, 1H), 8.10 (d, J = 8.1 Hz, 1H), 7.42 (dt, J = 8.3, 1.0 Hz, 1H), 7.28 (ddd, J = 8.3, 6.9, 1.2 Hz, 1H), 7.08 (ddd, J = 8.1, 6.9, 1.1 Hz, 1H), 4.33 (q, J = 7.1 Hz, 2H), 3.77 (t, J = 5.2 Hz, 4H), 2.57–2.51 (m, 4H), 2.27 (s, 3H), and 1.35 (t, J = 7.1 Hz, 3H); ^13^C NMR (75 MHz, DMSO-*d*_6_) δ 161.3, 135.8, 132.0, 125.3, 123.5, 120.9, 119.9, 118.7, 112.4, 60.2, 53.8, ~47.1 (br), 45.6, and 14.2; HRMS (ESI, *m*/*z*): calcd for C_16_H_22_N_5_O_2_+ [M + H]+: 316.1773, found: 316.1783.

BX-SI003: Ethyl (E)-3-(3-benzyl-3-methyltriaz-1-en-1-yl)-1H-indole-2-carboxylate; yield 37%; mp: 147 °C (from DMF-H_2_O); ^1^H NMR (300 MHz, DMSO-*d*_6_) δ 11.55 (s, 1H), 8.14 (d, J = 8.1 Hz, 1H), 7.47–7.23 (m, 7H), 7.06 (t, J = 7.6 Hz, 1H), 5.04 (s, 2H), 4.31 (q, J = 7.1 Hz, 2H), 3.19 (s, 3H), and 1.38–1.23 (m, 3H); ^13^C NMR (75 MHz, DMSO-*d*_6_) δ 161.5, 137.1, 135.9, 132.6, 128.6, 128.0, 127.4, 125.3, 123.6, 120.7, 119.2, 119.0, 112.4, 60.1, 58.7, 34.2, and 14.2; HRMS (ESI, *m*/*z*): calcd for C_19_H_21_N_4_O_2_+ [M + H]+: 337.1664, found: 337.1673.

BX-SI005: Ethyl (E)-3-((4-methyl-1,4-diazepan-1-yl)diazenyl)-1H-indole-2-carboxylate; yield 33%; mp: 110 °C (from DMF-H_2_O); ^1^H NMR (300 MHz, DMSO-*d*_6_) δ 11.45 (s, 1H), 8.10 (d, J = 8.1 Hz, 1H), 7.40 (d, J = 8.3 Hz, 1H), 7.26 (t, J = 7.6 Hz, 1H), 7.05 (t, J = 7.6 Hz, 1H), 4.31 (q, J = 7.1 Hz, 2H), 4.16–4.10 (m, 2H), 3.89–3.76 (m, 2H), 2.82–2.54 (m, 3H), 2.30 (s, 4H), 2.05–1.77 (m, 2H), and 1.34 (t, J = 7.1 Hz, 3H); ^13^C NMR (75 MHz, DMSO-*d*_6_) δ 161.5, 135.9, 133.2, 125.2, 123.6, 120.4, 119.1, 118.6, 112.3, 60.0, 58.7, 58.3, 53.5, 47.8, 46.0, 24.6, and 14.3; HRMS (ESI, *m*/*z*): calcd for C_17_H_24_N_5_O_2_+ [M + H]+: 330.1930, found: 330.1940.

BX-SI010: Ethyl (E)-3-(3-(2-(dimethylamino)ethyl)-3-methyltriaz-1-en-1-yl)-1H-indole-2-carboxylate; yield 19%; mp: 118 °C (from DMF-H_2_O); ^1^H NMR (300 MHz, DMSO-*d*_6_) δ 11.48 (s, 1H), 8.11 (d, J = 8.1 Hz, 1H), 7.39 (d, J = 8.4 Hz, 1H), 7.25 (t, J = 5.9 Hz, 1H), 7.05 (t, J = 7.2 Hz, 1H), 4.31 (q, J = 6.9 Hz, 2H), 3.90 (t, J = 5.7 Hz, 2H), 3.22 (s, br, 2H), 2.60–2.51 (m, 3H), 2.20 (s, 6H), and 1.34 (t, J = 6.6 Hz, 3H); ^13^C NMR (75 MHz, DMSO-*d*_6_) δ 161.6, 135.9, 133.0, 125.2, 123.6, 120.4, 119.1, 118.7, 112.3, 60.0, 57.3 (br), 53.4 (br), 45.2, 34.0, and 14.3; HRMS (ESI, *m*/*z*): calcd for C_16_H_24_N_5_O_2_+ [M + H]+: 318.1930, found: 318.1939.

BX-SI016: Ethyl (E)-3-((4-(2-methoxyethyl)piperidin-1-yl)diazenyl)-1H-indole-2-carboxylate; yield 28%; mp: 140 °C (from DMF-H_2_O); ^1^H NMR (300 MHz, DMSO-*d*_6_) δ 11.57 (s, 1H), 8.11 (d, J = 8.1 Hz, 1H), 7.41 (d, J = 8.3 Hz, 1H), 7.27 (t, J = 7.1 Hz, 1H), 7.06 (t, J = 7.8 Hz, 1H), 4.51–4.45 (m, 2H), 4.32 (q, J = 7.1 Hz, 2H), 3.39 (t, J = 6.4 Hz, 2H), 3.24 (s, 3H), 3.14–3.08 (m, 2H), 1.82 (d, J = 13.0 Hz, 2H), 1.73–1.67 (m, 1H), 1.50 (q, J = 6.5 Hz, 2H), and 1.40–1.19 (m, 5H); ^13^C NMR (75 MHz, DMSO-*d*_6_) δ 161.4, 135.9, 132.5, 125.3, 123.6, 120.7, 119.4, 118.8, 112.4, 69.6, 60.1, 57.9, 35.3, 32.4, 31.0, 14.2, and 11.9; HRMS (ESI, *m*/*z*): calcd for C_19_H_27_N_4_O_3_+ [M + H]+: 359.2083, found: 359.2093.

BX-SI055: Ethyl (E)-3-((4-methylpiperidin-1-yl)diazenyl)-1H-indole-2-carboxylate; yield 28%; mp: 177 °C (from DMF-H_2_O); ^1^H NMR (300 MHz, DMSO-*d*_6_) δ 11.57 (s, 1H), 8.10 (d, J = 8.1 Hz, 1H), 7.41 (d, J = 8.7 Hz, 1H), 7.27 (t, J = 7.5 Hz, 1H), 7.07 (t, J = 7.5 Hz, 1H), 4.51–4.45 (m, 2H), 4.32 (q, J = 7.1 Hz, 2H), 3.15–3.09 (m, 2H), 1.83–1.67 (m, 3H), 1.36 (t, J = 7.1 Hz, 3H), 1.36–1.21 (m, 2H), and 0.97 (d, J = 6.3 Hz, 3H); ^13^C NMR (75 MHz, DMSO-*d*_6_) δ 161.4, 135.9, 132.5, 125.2, 123.6, 120.7, 119.4, 118.8, 112.4, 60.1, 32.9, 30.3, 21.4, 14.2, and 11.2; HRMS (ESI, *m*/*z*): calcd for C_17_H_23_N_4_O_2_+ [M + H]+: 315.1821, found: 315.1830.

#### 4.1.3. Synthesis of Compounds BX-SI019, 020, 021, 027

A total of 0.22 g (0.001 mol) of Precursor 2, 5 mL of water, and 5 mL of dimethylformamide were placed in a round-bottom flask with a volume of 50 mL. The reaction mixture was cooled to 5 °C using an ice bath. Then, 0.3 mL of concentrated hydrochloric acid was added. After 15 min, the reaction mixture was treated with 0.086 g (0.00125 mol) of sodium nitrite and stirred at this temperature for 30 min. After that, 0.01 mol of the corresponding secondary amine 35–48 was added to the mixture, and it was heated at 50° C for 1 h. After cooling, the reaction mixture was diluted with 25 mL of water. The formed precipitate was filtered and dried.

BX-SI019: Ethyl (E)-3-(3,3-diethyltriaz-1-en-1-yl)-1-methyl-1H-pyrrolo[2,3-b]pyridine-2-carboxylate; yield 41%; oil; ^1^H NMR (300 MHz, DMSO-*d*_6_) δ 8.46 (q, J = 1.7 Hz, 1H), 8.44 (s, 1H), 7.23–7.13 (m, 1H), 4.35 (q, J = 7.1 Hz, 2H), 3.98 (s, 3H), 3.80 (q, J = 7.1 Hz, 4H), 1.35 (t, J = 7.1 Hz, 3H), and 1.30–1.15 (m, 6H); ^13^C NMR (75 MHz, DMSO-*d*_6_) δ 161.9, 147.1, 146.9, 132.7, 132.3, 120.0, 117.3, 111.0, 60.4, ~48.1 (br), 29.9, 14.1, and ~11.1 (br); HRMS (ESI, *m*/*z*): calcd for C_15_H_22_N_5_O_2_+ [M + H]+: 304.1773, found: 304.1779.

BX-SI020: Ethyl (E)-3-(3-benzyl-3-methyltriaz-1-en-1-yl)-1-methyl-1H-pyrrolo[2,3-b]pyridine-2-carboxylate; yield 39%; mp: 69 °C (from DMF-H_2_O); %; ^1^H NMR (300 MHz, DMSO-*d*_6_) δ 8.50 (dd, J = 7.9, 1.7 Hz, 1H), 8.47 (dd, J = 4.6, 1.7 Hz, 1H), 7.44–7.26 (m, 5H), 7.18 (dd, J = 8.0, 4.6 Hz, 1H), 5.05 (s, 2H), 4.33 (q, J = 7.1 Hz, 2H), 3.99 (s, 3H), 3.15 (s, 2H), and 1.29 (s, 4H); ^13^C NMR (75 MHz, DMSO-*d*_6_) δ 161.8, 147.1, 147.1, 137.0, 132.7, 131.7, 128.7, 128.0, 127.6, 120.6, 117.5, 111.0, 60.5, 58.8, 34.3, 30.0, and 14.1; HRMS (ESI, *m*/*z*): calcd for C_19_H_22_N_5_O_2_+ [M + H]+: 352.1773, found: 352.1782.

BX-SI021: Ethyl (E)-3-((4-benzylpiperazin-1-yl)diazenyl)-1-methyl-1H-pyrrolo[2,3-b]pyridine-2-carboxylate; yield 44%; mp: 78 °C (from DMF-H_2_O); ^1^H NMR (300 MHz, DMSO-*d*_6_) δ 8.47 (qd, J = 6.4, 1.6 Hz, 2H) 7.36 (d, J = 4.4 Hz, 4H), 7.33–7.23 (m, 1H), 7.23–7.14 (m, 1H), 4.33 (q, J = 7.1 Hz, 2H), 3.98 (s, 3H), 3.79 (t, J = 5.2 Hz, 4H), 3.58 (s, 2H), 2.57 (t, J = 5.2 Hz, 4H), and 1.32 (t, J = 7.1 Hz, 3H); ^13^C NMR (75 MHz, DMSO-*d*_6_) δ 161.6, 147.04, 147.03, 137.9, 132.6, 131.0, 128.9, 128.2, 127.1, 121.3, 117.6, 110.7, 61.7, 60.6, 51.7, 51.6, 30.0, and 14.1; HRMS (ESI, *m*/*z*): calcd for C_22_H_27_N_6_O_2_+ [M + H]+: 407.2195, found: 407.2205.

BX-SI027: Ethyl (E)-3-(3-(2-(dimethylamino)ethyl)-3-methyltriaz-1-en-1-yl)-1-methyl-1H-pyrrolo[2,3-b]pyridine-2-carboxylate; yield 41%; oil; ^1^H NMR (300 MHz, DMSO-*d*_6_) δ 8.49 (dd, J = 8.0, 1.7 Hz, 1H), 8.45 (dd, J = 4.6, 1.7 Hz, 1H), 7.18 (dd, J = 8.0, 4.6 Hz, 1H), 4.34 (q, J = 7.1 Hz, 2H), 3.98 (s, 3H), 3.93 (t, J = 6.3 Hz, 2H), 3.39–3.18 (m, 3H), 2.57 (t, J = 6.4 Hz, 2H), 2.21 (s, 6H), and 1.35 (t, J = 7.1 Hz, 3H); ^13^C NMR (75 MHz, DMSO-*d*_6_) δ 161.8, 147.1, 146.9, 132.7, 132.1, 120.1, 117.3, 111.0, 60.4, 57.2 (br), 52.9 (br), 45.2, 34.0, 29.9, and 14.1; HRMS (ESI, *m*/*z*): calcd for C_16_H_25_N_6_O_2_+ [M + H]+: 333.2038, found: 333.2048.

#### 4.1.4. Synthesis of Compounds BX-SI035, 036, 037, 038, 039, 040, 043, 044, 045, 048

A total of 0.22 g (0.001 mol) of Precursor 3, 5 mL of water, and 5 mL of dimethylformamide were placed in a round-bottom flask with a volume of 50 mL. The reaction mixture was cooled to 5 °C using an ice bath. Then, 0.3 mL of concentrated hydrochloric acid was added. After 15 min, the reaction mixture was treated with 0.086 g (0.00125 mol) of sodium nitrite and stirred at this temperature for 30 min. After that, 0.01 mol of the corresponding secondary amine 35–48 was added to the mixture, and it was heated at 50° C for 1 h. After cooling, the reaction mixture was diluted with 25 mL of water. The formed precipitate was filtered and dried.

BX-SI035: Ethyl ester ethyl (E)-6-fluoro-3-((4-methylpiperazin-1-yl)diazenyl)-1H-indole-2-carboxylate; yield 23%; mp: 186 °C (from DMF-H_2_O); ^1^H NMR (300 MHz, DMSO-*d*_6_) δ 11.71 (s, 1H), 8.11 (dd, J = 9.0, 5.8 Hz, 1H), 7.12 (dd, J = 9.9, 2.4 Hz, 1H), 6.95 (td, J = 9.4, 2.4 Hz, 1H), 4.32 (q, J = 7.0 Hz, 2H), 3.78 (t, J = 5.2 Hz, 4H), 2.52 (s, 3H), 2.27 (m, 4H), and 1.34 (t, J = 7.1 Hz, 3H); ^13^C NMR (75 MHz, DMSO-*d*_6_) δ 161.0, 160.8 (d, J = 240.0 Hz), 136.0 (d, J = 13.8 Hz), 132.2, 125.4 (d, J = 10.0 Hz), 120.4 (d, J = 2.9 Hz), 115.8, 109.9 (d, J = 24.7 Hz), 97.9 (d, J = 25.5 Hz), 60.2, 53.8, ~47.1 (br), 45.6, 39.5, and 14.2; HRMS (ESI, *m*/*z*): calcd for C_16_H_21_FN_5_O_2_+ [M + H]+: 334.1679, found: 334.1689.

BX-SI036: Ethyl (E)-3-(3,3-diethyltriaz-1-en-1-yl)-6-fluoro-1H-indole-2-carboxylate; yield 29%; mp: 137 °C (from DMF-H_2_O); ^1^H NMR (300 MHz, DMSO-*d*_6_) δ 11.53 (s, 1H), 8.08 (dd, J = 9.0, 5.8 Hz, 1H), 7.09 (dd, J = 9.9, 2.4 Hz, 1H), 6.92 (ddd, J = 9.6, 9.0, 2.4 Hz, 1H), 4.32 (q, J = 7.1 Hz, 2H), 3.79 (q, J = 7.1 Hz, 4H), 1.34 (t, J = 7.1 Hz, 3H), and 1.26 (t, J = 6.9 Hz, 6H); ^13^C NMR (75 MHz, DMSO-*d*_6_) δ 161.4, 160.8 (d, J = 240.0 Hz), 159.2, 136.2 (d, J = 12.9 Hz), 133.3, 125.4 (d, J = 10.2 Hz), 119.3 (d, J = 3.3 Hz), 116.1, 109.5 (d, J = 24.4 Hz), 97.6 (d, J = 25.4 Hz), 60.1, ~48.3 (br), 14.2, and ~11.3 (br); HRMS (ESI, *m*/*z*): calcd for C_15_H_20_FN_4_O_2_+ [M + H]+: 307.1570, found: 307.1581.

BX-SI037: Ethyl (E)-3-(3-benzyl-3-methyltriaz-1-en-1-yl)-6-fluoro-1H-indole-2-carboxylate; yield 37%; mp: 162 °C (from DMF-H_2_O); ^1^H NMR (300 MHz, DMSO-*d*_6_) δ 11.62 (s, 1H), 8.14 (dd, J = 9.0, 5.8 Hz, 1H), 7.44–7.26 (m, 5H), 7.12 (dd, J = 10.1, 2.4 Hz, 1H), 6.93 (td, J = 9.7, 2.5 Hz, 1H), 5.04 (s, 2H), 4.31 (q, J = 7.1 Hz, 2H), 3.18 (s, 3H), and 1.31 (t, J = 7.1 Hz, 3H); ^13^C NMR (75 MHz, DMSO-*d*_6_) δ 162.4, 161.2, 160.8 (d, J = 240.0 Hz), 136.1 (d, J = 13.1 Hz), 132.8, 128.6, 128.0, 127.5, 125.5 (d, J = 10.1 Hz), 119.7 (d, J = 3.6 Hz), 116.0, 109.6 (d, J = 24.3 Hz), 97.7 (d, J = 25.9 Hz), 60.1, 58.3, 33.6, and 14.2; HRMS (ESI, *m*/*z*): calcd for C_19_H_20_FN_4_O_2_+ [M + H]+: 355.1570, found: 355.1580.

BX-SI038: Ethyl (E)-3-((4-benzylpiperazin-1-yl)diazenyl)-6-fluoro-1H-indole-2-carboxylate; yield 44%; mp: 171 °C (from DMF-H_2_O); δ 11.71 (s, 1H), 8.11 (dd, J = 9.0, 5.8 Hz, 1H), 7.40–7.22 (m, 5H), 7.12 (dd, J = 9.9, 2.4 Hz, 1H), 6.94 (ddd, J = 9.6, 8.9, 2.4 Hz, 1H), 4.31 (q, J = 7.1 Hz, 2H), 3.79 (t, J = 5.2 Hz, 4H), 3.58 (d, J = 2.7 Hz, 2H), 2.58 (t, J = 5.2 Hz, 4H), and 1.32 (t, J = 7.1 Hz, 3H); ^13^C NMR (75 MHz, DMSO-*d*_6_) δ 161.1, 160.8 (d, J = 240.0 Hz), 137.9, 136.1 (d, J = 13.0 Hz), 132.2, 128.9, 128.3, 127.1, 125.5 (d, J = 10.7 Hz), 120.4 (d, J = 3.4 Hz), 115.8, 109.9 (d, J = 24.23 Hz), 97.8 (d, J = 25.3 Hz), 61.7, 60.2, 51.8, and 14.2; HRMS (ESI, *m*/*z*): calcd for C_19_H_20_FN_4_O_2_+ [M + H]+: 355.1570, found: 355.1580.

BX-SI039: Ethyl (E)-6-fluoro-3-((4-methyl-1,4-diazepan-1-yl)diazenyl)-1H-indole-2-carboxylate; yield 39%; mp: 176 °C (from DMF-H_2_O); ^1^H NMR (300 MHz, DMSO-*d*_6_) δ 11.53 (s, 1H), 8.11 (dd, J = 9.0, 5.8 Hz, 1H), 7.10 (dd, J = 9.9, 2.4 Hz, 1H), 6.92 (td, J = 9.4, 2.4 Hz, 1H), 4.31 (q, J = 7.1 Hz, 2H), 4.11 (d, J = 17.4 Hz, 2H), 3.78 (d, J = 17.0 Hz, 2H), 2.82–2.53 (m, 4H), 2.30 (s, 3H), 2.05–1.78 (m, 2H), and 1.33 (t, J = 7.1 Hz, 3H); ^13^C NMR (75 MHz, DMSO-*d*_6_) δ 161.2, 160.9 (d, J = 240.0 Hz), 136.1 (d, J = 12.7 Hz), 133.3, 125.5 (d, J = 10.3 Hz), 119.1, 116.1, 109.4 (d, J = 24.6 Hz), 97.6 (d, J = 25.5 Hz), 58.7, 58.7, 58.3, 53.6, 47.9, 46.0, 24.5, and 14.3; HRMS (ESI, *m*/*z*): calcd for C_17_H_23_FN_5_O_2_+ [M + H]+: 348.1835, found: 348.1845.

BX-SI040: Ethyl (E)-6-fluoro-3-(morpholinodiazenyl)-1H-indole-2-carboxylate; yield 39%; mp: 182 °C (from DMF-H_2_O); ^1^H NMR (300 MHz, DMSO-*d*_6_) δ 11.76 (s, 1H), 8.12 (dd, J = 9.0, 5.8 Hz, 1H), 7.13 (dd, J = 9.8, 2.4 Hz, 1H), 6.96 (ddd, J = 9.7, 8.9, 2.4 Hz, 1H), 4.33 (q, J = 7.1 Hz, 2H), 3.86–3.72 (m, 8H), and 1.34 (t, J = 7.1 Hz, 3H); ^13^C NMR (75 MHz, DMSO-*d*_6_) δ 161.0, 160.8 (d, J = 240.0 Hz), 136.0 (d, J = 13.8 Hz), 132.2, 125.4 (d, J = 10.0 Hz), 120.7 (d, J = 3.2 Hz), 115.8, 110.0 (d, J = 24.4 Hz), 97.9 (d, J = 25.8 Hz), 65.6, 60.3, 47.6, 39.5, and 14.2; HRMS (ESI, *m*/*z*): calcd for C_15_H_18_FN_4_O_3_+ [M + H]+: 321.1362, found: 321.1374.

BX-SI043: Ethyl (E)-6-fluoro-3-(pyrrolidin-1-yldiazenyl)-1H-indole-2-carboxylate; yield 39%; mp: 193 °C (from DMF-H_2_O); ^1^H NMR (300 MHz, DMSO-*d*_6_) 11.51 (s, 1H), 8.08 (dd, J = 9.0, 5.8 Hz, 1H), 7.09 (dd, J = 10.0, 2.4 Hz, 1H), 6.92 (td, J = 9.3, 2.5 Hz, 1H), 4.30 (q, J = 7.1 Hz, 2H), 3.77 (s, br, 4H), 2.00 (t, J = 6.7 Hz, 4H), and 1.34 (t, J = 7.1 Hz, 3H); ^13^C NMR (75 MHz, DMSO-*d*_6_) δ 161.8, 160.7 (d, J = 240.0 Hz), 136.2 (d, J = 12.8 Hz), 134.2, 125.9 (d, J = 10.4 Hz), 119.5 (d, J = 3.2 Hz), 116.6, 109.7 (d, J = 24.7 Hz), 98.0 (d, J = 25.4 Hz), 60.5, 23.8, 14.7, and 10.2; HRMS (ESI, *m*/*z*): calcd for C_15_H_18_FN_4_O_2_+ [M + H]+: 305.1413, found: 305.1424.

BX-SI044: Ethyl (E)-3-(3-(2-(dimethylamino)ethyl)-3-methyltriaz-1-en-1-yl)-6-fluoro-1H-indole-2-carboxylate; yield 27%; mp: 172 °C (from DMF-H_2_O); ^1^H NMR (300 MHz, DMSO-*d*_6_) δ 11.54 (s, 1H), 8.13 (dd, J = 9.0, 5.8 Hz, 1H), 7.10 (dd, J = 9.9, 2.4 Hz, 1H), 6.93 (td, J = 9.3, 2.4 Hz, 1H), 4.31 (q, J = 7.1 Hz, 2H), 3.91 (t, J = 6.3 Hz, 2H), 3.22 (s, br, 4H), 2.57 (t, J = 6.4 Hz, 2H), 2.21 (s, 6H), and 1.34 (t, J = 7.1 Hz, 3H); ^13^C NMR (75 MHz, DMSO-*d*_6_) δ 161.3, 160.8 (d, J = 240.0 Hz), 136.0 (d, J = 13.1 Hz), 133.2, 125.4 (d, J = 10.4 Hz), 119.2 (d, J = 3.6 Hz), 116.1, 109.6 (d, J = 24.5 Hz), 97.6 (d, J = 25.3 Hz), 60.0, 57.3, 52.8, 45.2, 33.9, and 14.2; HRMS (ESI, *m*/*z*): calcd for C_16_H_23_FN_5_O_2_+ [M + H]+: 336.1835, found: 336.1843.

BX-SI045: Ethyl (E)-6-fluoro-3-((4-methylpiperidin-1-yl)diazenyl)-1H-indole-2-carboxylate; yield 36%; mp: 183 °C (from DMF-H_2_O); ^1^H NMR (300 MHz, DMSO-*d*_6_) δ 11.63 (s, 1H), 8.12 (dd, J = 8.9, 5.8 Hz, 1H), 7.11 (dd, J = 9.9, 2.4 Hz, 1H), 6.93 (td, J = 9.3, 2.4 Hz, 1H), 4.48 (s, 2H), 4.32 (q, J = 7.1 Hz, 2H), 3.14 (s, 2H), 1.82–1.64 (m, 3H), 1.39–1.20 (m, 5H), and 0.96 (d, J = 6.2 Hz, 3H); ^13^C NMR (75 MHz, DMSO-*d*_6_) δ 161.1, 160.8 (d, J = 240.0 Hz), 136.0 (d, J = 13.1 Hz), 132.7, 125.5 (d, J = 10.1 Hz), 119.8 (d, J = 3.3 Hz), 115.9, 109.5 (d, J = 24.6 Hz), 97.6 (d, J = 26.3 Hz), 60.1, 32.9, 30.3, 21.4, 14.2, and 11.2; HRMS (ESI, *m*/*z*): calcd for C_17_H_22_FN_4_O_2_+ [M + H]+: 333.1726, found: 333.1736.

BX-SI048: Ethyl (E)-6-fluoro-3-(3-methyl-3-(2-(pyridin-2-yl)ethyl)triaz-1-en-1-yl)-1H-indole-2-carboxylate; yield 36%; mp: 76 °C (from DMF-H_2_O); ^1^H NMR (300 MHz, DMSO-*d*_6_) δ 11.53 (s, 1H), 8.58–8.50 (m, 1H), 7.89–7.63 (m, 2H), 7.33 (d, J = 7.8 Hz, 1H), 7.21 (t, J = 6.4 Hz, 1H), 7.12–7.02 (m, 1H), 6.88 (t, J = 9.4 Hz, 1H), 4.30 (q, J = 7.0 Hz, 2H), 4.24–4.18 (m, 2H), 3.26–3.12 (m, 5H), and 1.32 (t, J = 7.0 Hz, 3H); ^13^C NMR (75 MHz, DMSO-*d*_6_) δ 161.3, 160.8 (d, J = 240.0 Hz), 149.1, 136.5 (d, J = 12.5 Hz), 135.9, 133.0, 125.4 (d, J = 10.1 Hz), 123.5, 121.6, 119.3 (d, J = 3.4 Hz), 119.2, 116.0, 109.2 (d, J = 24.2 Hz), 97.4 (d, J = 26.6 Hz), 60.0, 55.3, 36.5, 33.9, and 14.2; HRMS (ESI, *m*/*z*): calcd for C_19_H_21_FN_5_O_2_+ [M + H]+: 370.1679, found: 370.1690.

#### 4.1.5. Synthesis of Compounds BX-SI057, 058

A total of 0.22 g (0.001 mol) of Precursor 4, 5 mL of water, and 5 mL of dimethylformamide were placed in a round-bottom flask with a volume of 50 mL. The reaction mixture was cooled to 5 °C using an ice bath. Then, 0.3 mL of concentrated hydrochloric acid was added. After 15 min, the reaction mixture was treated with 0.086 g (0.00125 mol) of sodium nitrite and stirred at this temperature for 30 min. After that, 0.01 mol of the corresponding secondary amine 35–48 was added to the mixture, and it was heated at 50° C for 1 h. After cooling, the reaction mixture was diluted with 25 mL of water. The formed precipitate was filtered and dried.

BX-SI057: Ethyl (E)-3-((4-methylpiperazin-1-yl)diazenyl)-1H-pyrrolo[2,3-b]pyridine-2-carboxylate; yield 33%; mp: 180 °C (from DMF-H_2_O); ^1^H NMR (300 MHz, DMSO-*d*_6_) δ 12.14 (s, 1H), 8.46 (dd, J = 8.0, 1.7 Hz, 1H), 8.41 (dd, J = 4.6, 1.7 Hz, 1H), 7.15 (dd, J = 8.1, 4.6 Hz, 1H), 4.33 (q, J = 7.1 Hz, 2H), 3.80 (t, J = 5.2 Hz, 4H), 2.55–2.51 (m, 4H), 2.27 (s, 3H), and 1.34 (t, J = 7.1 Hz, 3H); ^13^C NMR (75 MHz, DMSO-*d*_6_) δ 161.0, 147.4, 147.0, 132.9, 132.5, 119.6, 117.4, 111.5, 60.4, 53.5, 45.6, and 14.2; HRMS (ESI, *m*/*z*): calcd for C_15_H_21_N_6_O_2_+ [M + H]+: 317.1725, found: 317.1736.

BX-SI058: Ethyl (E)-3-((4-methylpiperidin-1-yl)diazenyl)-1H-pyrrolo[2,3-b]pyridine-2-carboxylate; yield 35%; mp: 240 °C (from DMF-H_2_O); ^1^H NMR (300 MHz, DMSO-*d*_6_) δ 1H NMR (300 MHz, DMSO-d6) δ 12.09 (s, 1H), 8.46 (dd, J = 8.0, 1.7 Hz, 1H), 8.40 (dd, J = 4.6, 1.7 Hz, 1H), 7.14 (dd, J = 8.0, 4.6 Hz, 1H), 4.52–4.46 (m, 2H), 4.32 (q, J = 7.1 Hz, 2H), 3.19–3.13 (m, 2H), 1.79 (d, J = 13.1 Hz, 3H), 1.34 (t, J = 6.9 Hz, 3H), 1.30–1.20 (m, 2H), and 0.98 (d, J = 6.2 Hz, 3H); HRMS (ESI, *m*/*z*): calcd for C_16_H_22_N_5_O_2_+ [M + H]+: 316.1773, found: 316.1784.

### 4.2. Determination of Solubility of Compounds in Water

The solubility of the compounds in water was assessed using HPLC analysis.

Two 1.0 mg portions of the test compounds were dissolved in 10 μL of dimethyl sulfoxide; 1.0 mL of acetonitrile was added to one sample (standard), and 1.0 mL of water was added to the second sample (tested sample). Both vials were mixed in a shaker for 5 min, then centrifuged at 10,000 rpm for 5 min. The supernatants were collected and analyzed using Agilent 1290 Infinity (Agilent Technologies, Santa Clara, CA, USA) with a YMC-Triart C18 50 × 2.0 mm, 1.9 μm column. The concentration of the test compound in water was determined by comparing the peak areas of the standard (concentration 1.0 mg/mL) and the tested sample.

### 4.3. Determination of the Minimum Inhibitory Concentrations (MICs)

Clinical MRSA isolates with multidrug resistance were obtained from the collection of the Research Institute of Antimicrobial Chemotherapy, Smolensk State Medical Academy. Each strain exhibited an antibiotic susceptibility profile, determined according to the EUCAST breakpoints [41].

The MICs were determined using a broth microdilution method according to [42]. Briefly, serial dilutions of the tested compounds in 50 µL of Mueller Hinton broth (BD) were prepared in 96-well microplates. The antibiotics vancomycin and linezolid (Sigma-Aldrich, St. Louis, MO, USA) were used as a positive control. Bacterial suspensions of the tested MRSA strains were prepared in the Mueller Hinton broth (BD) with a concentration 1 × 10^6^ CFU/mL, and then 50 µL aliquots were added to the prepared microplates. The final concentrations of the antibiotics in the wells were 1, 0.5, 0.25, and 0.125 mg/L, for the tested compounds, and 2 mg/L for vancomycin and linezolid. Negative control (lack of bacterial cells) and growth control (lack of compound) wells were included in all plates. The microplates were incubated at 37 °C for 18 h. The growth was assessed visually after incubation, and the MIC was assumed as the lowest compound concentration at which a noticeable growth of the microorganisms was inhibited (no visible growth). All experiments were conducted in triplicate.

### 4.4. Cytotoxicity Study

The primary screening of 22 substances was performed on human embryonic fibroblasts (HEFs). The hepatotoxicity of the six substances most active against MRSA was additionally studied on HepG2 human liver carcinoma cells. The HepG2 cell line was obtained from the American Type Culture Collection (ATCC cat. # HB-8065). The primary HEF cells (normal human embryonic fibroblasts) were derived from human embryonic stem cells. The cells were cultured in DMEM/F12 (1:1) medium containing 10% fetal bovine serum (Invitrogen, Carlsbad, CA, USA) at 37 °C and 5% CO_2_, according to standard mammalian tissue culture protocols and sterile technique. All cell lines were tested before use using a LookOut^®^ Mycoplasma PCR Detection Kit (Sigma-Aldrich, USA), according to the manufacturer’s protocol, and were found to be free of Mycoplasma infection.

The incubation time with the substance was 24 h for all substances and cell lines. Cell detachment by trypsin, centrifugation, and counting in a hemocytometer were performed 16 h before the addition of the substance. For correct counting of the HepG2 cells, the cell suspension was passed through a syringe needle to get rid of cell aggregates. The cells were placed in the wells of a 96-well plate. The number of cells was selected according to the properties of the cell line and the final optical density in the experiment. Fibroblasts and hepatocytes were used in the amount of 10 thousand cells per well in 100 μL of DMEM/F12 medium supplemented with 10% fetal bovine serum. The substances were added to 100 µL of medium in the wells so that the final concentrations were 50, 25, 12.5, 6.25, 3.125, and 1.62 mg/L. The final well volume was 200 µL. The medium containing fetal bovine serum, without added substances, was used as a negative control. After 24 h, excessive amounts of medium were discarded. A total of 30 µL of MTT solution (5 mg/mL in PBS) was added to each well. After the precipitation of formazan crystals (3 h incubation), they were dissolved using 100 μL of dimethyl sulfoxide. The optical density (OD) was measured at 570 nm using a microplate reader (Eppendorf). For every condition, the cell viabilities were assessed in at least three independent experiments for every cell culture.

Cell survival was calculated using the following formula:(OD of treated cells − OD blank)/(OD of control cells − OD blank) × 100%,
where OD blank represents the OD values in wells without cells but with the addition of the MTT and DMSO solutions.

The results were presented in the form of cytotoxicity curves, and the graphs were also plotted in a Microsoft Excel program; the calculation of the IC50 half-maximal inhibition concentration values was performed in a GraphPad Prism 6.0 (GraphPad Software Inc., San Diego, CA, USA) program, based on the equations of the cytotoxicity curves.

### 4.5. UPLC–MS/MS Analysis

During ADME studies, the concentration of the compounds was analyzed using ultra-performance liquid chromatography (UPLC) coupled with tandem mass spectrometry (MS/MS) in the multiple reaction monitoring (MRM) mode. A quantitative analysis was performed using the internal standard (IS) calibration method, with tolbutamide as the IS.

The analyses were carried out on a system composed of an Agilent 1290 Infinity UPLC system (Agilent Technologies, USA) connected by ESI interface to a tandem quadrupole mass spectrometer QTRAP 6500 (AB Sciex Corp, Framingham, MA, USA). The separation was conducted on an YMC Triart C18 50 × 2 mm, 1.9 μm column, which was kept at 35 °C. The samples were cooled to 8 °C in an autosampler, and aliquots of 1 µL were delivered onto the column. Elution was performed using gradients of A (0.1% formic acid in water) and B (0.1% formic acid in acetonitrile), at the flow rate 0.35 mL/min. The gradient was used as follows: linear gradient of from 60 to 99% within 0.5 min, followed by isocratic 99% B for 1.6 min, then reconditioning to 60% A. The MS/MS analyses were performed in MRM mode and collected in the positive mode. The MS parameters were as follows: source temperature, 500 C; entrance potential, 10 v; ion spray voltage, 4500 V; collision gas pressure, 35 psi; nebulizing gas pressure, 50 psi; drying gas pressure, 50 psi. The declustering potential was 31 V for BX and 93 V for IS, resepectively; the collision energy was attuned to 17 eV; the collision cell exit potential was 10 V for BX and 8 V for IS MRM, respectively; the analyses were recorded for 305 → 234 (BX) and 271 → 172 (tolbutamide). Data acquisition was controlled using MassLynx v. 4.1 software (Waters Corp, Milford, MA, USA). The mass spectrum and the MRM chromatogram of compound BX are presented in Appendix A.

### 4.6. Evaluation of Stability in Artificial Stomach and Intestinal Juices

For artificial gastric juice, we used simulated gastric fluid (RICCA Chemical Company, Arlington, TX, USA), with the addition of pepsin to a final concentration of 6.4 mg/mL. The final solution had a pH of 1.2, which emulates the stomach environment in the fasting state.

The artificial intestinal juice was prepared by dissolving 6.8 g of KH_2_PO_4_ and 77 mL of 0.2N NaOH in 750 mL of water and adding pancreatin (Biosintez, Penza, Russia) for the final concentration of 10 mg/mL. The final pH of 6.8 was adjusted with the addition of water.

Chlorambucil (Sigma-Aldrich, USA), which is stable in gastric juice and unstable in intestinal juice, was used as a control compound.

For stability evaluation, 3 μL of 100X solutions of the tested or control compound (0.1 mM) was added to 297 μL of the corresponding juice (prewarmed to 37 °C) to a final concentration of 1 μM. The juice samples were incubated with the test and control compounds in the 0.65 mL microtubes at 37 °C and under 300 rpm rotation. After 15, 60, and 240 min, 30 μL aliquots were taken into 1.1 mL microtubes with 180 μL of chilled acetonitrile with an internal standard of 200 ng/mL of tolbutamide (Fluka, Buchs, Switzerland). The tubes were stirred and kept at 4 °C for 15 min, then centrifuged at 1500× *g* for 10 min. A total of 200 μL of supernatants were transferred into microtubes and stored at −70 °C until UPLC–MS/MS analysis.

### 4.7. Evaluation of Cellular Barrier Penetration In Vitro

The MultiScreen Caco-2 (ATCC cat. # HTB-37) test system (Millipore, St. Charles, MO, USA) was used to study the transport through a cell monolayer, according to the manufacturer’s instructions. Briefly, a study of transport from the apical membrane to the basolateral (A-B) area and in the opposite direction (B-A) was carried out on a 21-day culture of Caco-2 at a 1 µM concentration of the tested compound for 2 h. Monolayer integrity was verified by measuring electrical resistance (TEER) using a Millicell-ERS instrument. A TEER value of at least 3 KOhm/well was determined to be acceptable. Several control compounds, purchased from Sigma-Aldrich, were used, i.e., ranitidine (low permeability), propranolol (high permeability), and rhodamine (Pgp transport).

### 4.8. Determination of Stability in Liver Microsomes

The liver metabolism of the substance was studied in the microsomal fractions of human and rat livers (XenoTech HMMCPL, RTMCPL, Kansas City, KS, USA) in the presence of the cofactor of the oxidative enzymes NADPH (AppliChem, Darmstadt, Germany).

A total of 208 µL of 1.156X microsome solution in a phosphate buffer with pH 7.4 (BD Gentest) was placed into each well of a 96-well plate. The plate was incubated in a thermal shaker for 10 min at 400 rpm and 37 °C; then, 2.4 µL of 100× stocks of the tested substance was added into each well. A total of 30 µL of 8× cofactor solution was added to each experimental well (two replicates), 30 µL of deionized water was added to one control well, and all wells were mixed three times using a multichannel pipette with a volume of 100 µL. After mixing, the plate was placed in a thermal shaker at 400 rpm and 37 °C. The reactions were terminated at the following time points—0, 5, 10, 15, 20, and 30 min—by transferring 30 µL of the reaction mixture from the corresponding well to 1.1 mL microtubes and adding 180 µL of acetonitrile with internal standard 200 ng/mL tolbutamide (Fluka, Switzerland). To evaluate the stability of the substance in the microsomes without the cofactor, two aliquots were taken from the control well, and the reactions were terminated at 0 and 30 min time points. Protein precipitation was carried out on ice in a refrigerator at +4 °C for 15 min. Then, the samples were centrifuged for 10 min at 1500× *g*, and 150 μL of the supernatant was taken for UPLC–MS/MS analysis.

The metabolic parameters were calculated from the graph of the Ln normalized areas of the chromatographic peaks versus time, as follows:
k = −α;
where k is the elimination rate constant; α is the slope of the linear section of the graph
t_1/2_ = 0.693/k;
where t_1/2_ is the half-life time (min); k is the elimination rate constant
Cl_int_ = k × 1000/m;
where Cl_int_ is in vitro clearance (μL/min/mg protein); k is the elimination rate constant; m is the concentration of the microsomes (mg/mL).

### 4.9. Evaluation of Inhibition of Cytochrome P450 Isoforms Activities

The effect of the tested compound on the activity of the main isoforms of human liver cytochrome P450 (1A2, 2C9, 2C19, 2D6, 3A4, 2C8) was investigated using human liver microsomes (XenoTech, USA) and their corresponding substrates: phenacetin, testosterone, tolbutamide, S-mephenytoin, dextromethorphan hydrobromide, and amodiaquine dihydrochloride dihydrate (Sigma-Aldrich, USA). Well-known specific isoform inhibitors were used as controls: alpha-naphthoflavone, sulfaphenazole, fluvoxamine, quinidine, quercetin, and ketoconazole (Sigma-Aldrich).

Briefly, the tested compound at a concentration of 0.0046–10 μM was incubated with human microsomes in the presence of NADPH and a mixture of seven substrates. The reaction was stopped after 20 min by adding acetonitrile. The resulting metabolites were detected using UPLC–MS/MS. The calculation of the IC50 half-maximal inhibition concentration values was performed in the GraphPad Prism 6.0 (GraphPad Software Inc., USA) program, based on the equations of the experimental curves.

### 4.10. Evaluation of Stability in Rat and Human Plasma

For the experiment, human and rat plasma (SD) was thawed at room temperature and centrifuged for 5 min at 1500× *g*. Eucatropine (Sigma-Aldrich, USA), which is unstable in human and rat plasma, was used as a control compound.

For stability evaluation, 3 μL of 100X solutions of the tested or control compound (0.1 mM) was added to 297 μL of the corresponding plasma (prewarmed to 37 °C) to a final concentration of 1 μM. The plasma samples were incubated with the test and control compounds in the 0.65 mL microtubes at 37 °C and under 300 rpm rotation. After 15, 30, 60, and 240 min, 30 μL aliquots were taken into 1.1 mL microtubes with 180 μL of chilled acetonitrile with an internal standard of 200 ng/mL tolbutamide (Fluka). The tubes were stirred and kept at 4 °C for 15 min, then centrifuged at 1500× *g* for 10 min. A total of 200 μL of the supernatants was transferred into microtubes and stored at −70 °C until UPLC–MS/MS analysis.

### 4.11. Determination of Binding to Plasma Proteins

The binding of the tested substance to human and rat plasma proteins was studied using equilibrium dialysis. To determine protein binding, pooled human plasma samples diluted with buffer to 50% were used. The test was performed in a Teflon-coated 48-well dialysis plate. Each well contained two separate chambers separated by a vertical semi-permeable dialysis membrane with 8 kDa pores. A plasma sample with 1 μM of the test compound was introduced into one of the chambers, and a buffer solution with pH 7.2 was placed into the other chamber. Over time, passive diffusion of the unbound compound occurred, and a state of equilibrium was reached between the plasma and buffer chambers. The amount of free fraction was assessed using UPLC–MS/MS. During the study, the stability of the compounds for 4 h and their passive permeability through the dialysis membrane were also assessed using mass balance. Warfarin (Sigma-Aldrich, USA) was used as a control compound at a concentration of 1 µM.

### 4.12. Investigation of Potential Cardiotoxicity Due to hERG Blockage

Determination of the binding of the BX-SI043 compound to the hERG K^+^-ion channel was carried out using the Predictor hERG Fluorescence Polarization Assay Kit (Invitrogen, USA) at concentrations of 0.016, 0.08, 0.4, 2, 10, and 50 µM, according to the manufacturer’s instructions. This method is based on the measurement of the changing fluorescence polarization caused by the displacement of the high-affinity fluorescent tracer by the tested compound. A selective inhibitor of the hERG channels E-4031 was used as a control compound. Briefly, a 2X solution of membranes, containing hERG channels, was prepared and placed in 384-well plate. The tested substance, the control substance, the buffer, and the tracer were placed in the corresponding wells and incubated in the dark at room temperature for 2 h. Fluorescence polarization measurement was conducted on an Infinite M1000 PRO plate reader with the following parameters: 10 flashes per well; an excitation center wavelength of 530 nm, with a bandwidth of 5 nm; an emission center wavelength of 585 nm, with a bandwidth of 585 nm. The calculation of the IC50 half-maximal inhibition concentration values was performed in GraphPad Prism 6.0 (GraphPad Software Inc.), based on the equations of the experimental curves.

### 4.13. Animal Toxicology Studies (Single-Dose Acute Toxicity of Rats)

The experimental design and animal handling protocols were carried out in accordance with EU Directive 2010/63/EU for animal experiments.

A total of 56 Wistar outbred rats (aged 6–9 weeks, with bodyweights of 185–205 g) were purchased from RMC “HOME OF PHARMACY” and kept for one week under standard adaptive conditions, after which they were randomly divided between a control group (four male and four female) and four treatment groups (six male and six female).

Before administration, the BX-SI043 substance was resuspended in a 1% starch solution. The animals in the treatment groups received suspensions intragastrically in doses of 2000 mg/kg (administered in two equal doses, with a 30–40 min interval), 1000 mg/kg, 600 mg/kg, and 300 mg/kg (administered in single doses). The control group received intragastrically a 1% starch solution in a volume of 13.2 mL/kg (in two equal doses, with a 30–40 min interval).

The animals were monitored daily for any abnormalities. On the 2nd, 7th, and 13th days after administration, the animals were clinically evaluated and weighed, and the consumption of feed and water were measured. On the 14th day, the behavior of the animals was evaluated. On the 15th day, the animals were euthanized, and the lungs (with trachea), heart, thymus, liver, kidneys, spleen, stomach, small intestine, colon, brain, adrenal glands, and testes/ovaries were fixed in 10% neutral formalin for histological evaluation.

## Figures and Tables

**Figure 1 ijms-26-01870-f001:**
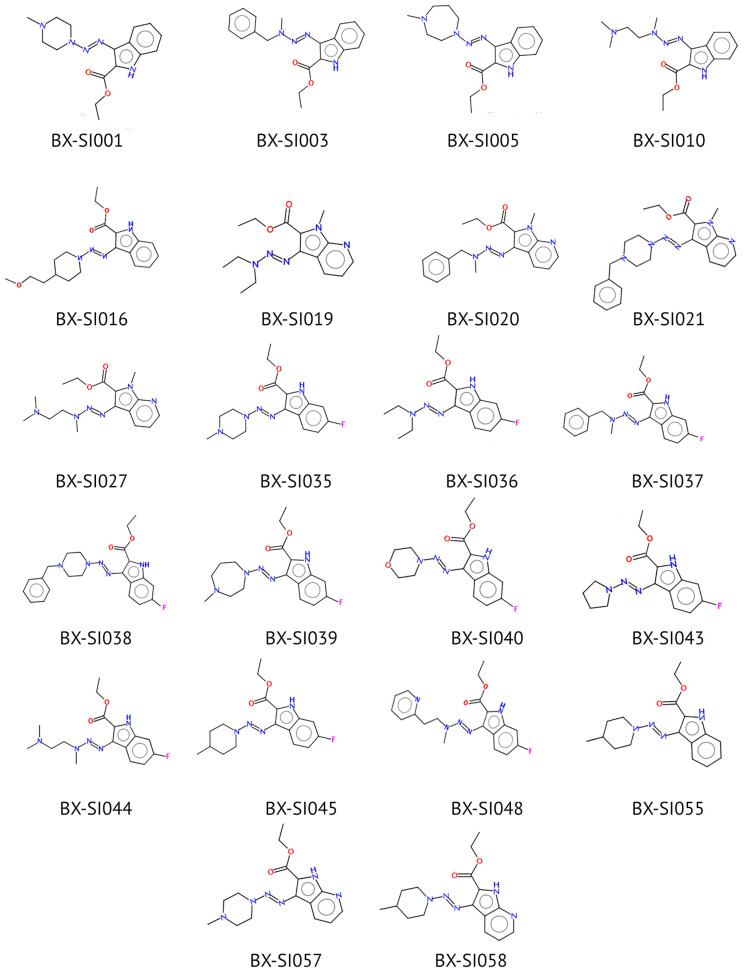
Structures of tested compounds.

**Figure 2 ijms-26-01870-f002:**
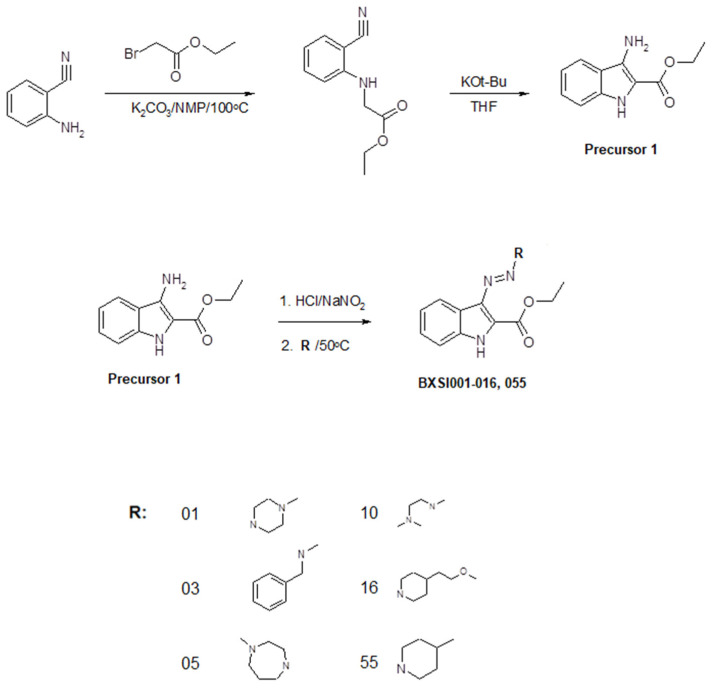
Synthesis of compounds BX-SI001, 003, 005, 010, 016, and 055.

**Figure 3 ijms-26-01870-f003:**
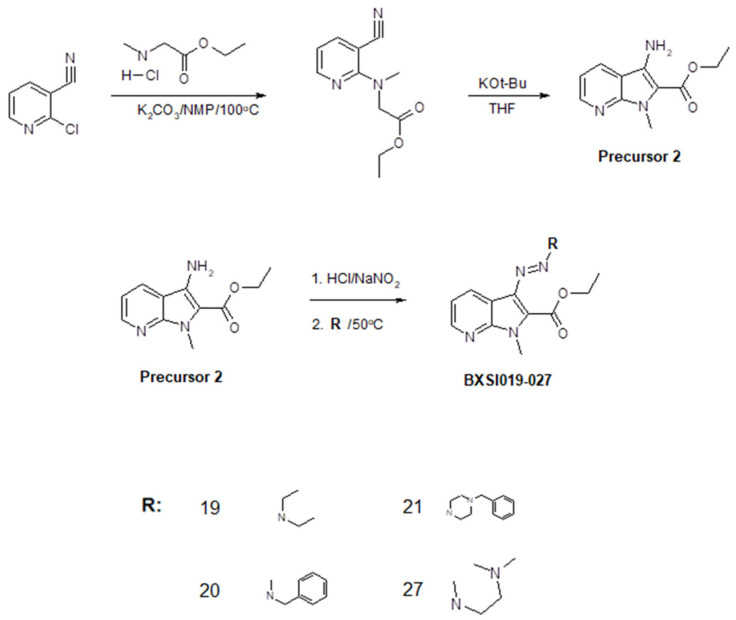
Synthesis of compounds BX-SI019, 020, 021, and 027.

**Figure 4 ijms-26-01870-f004:**
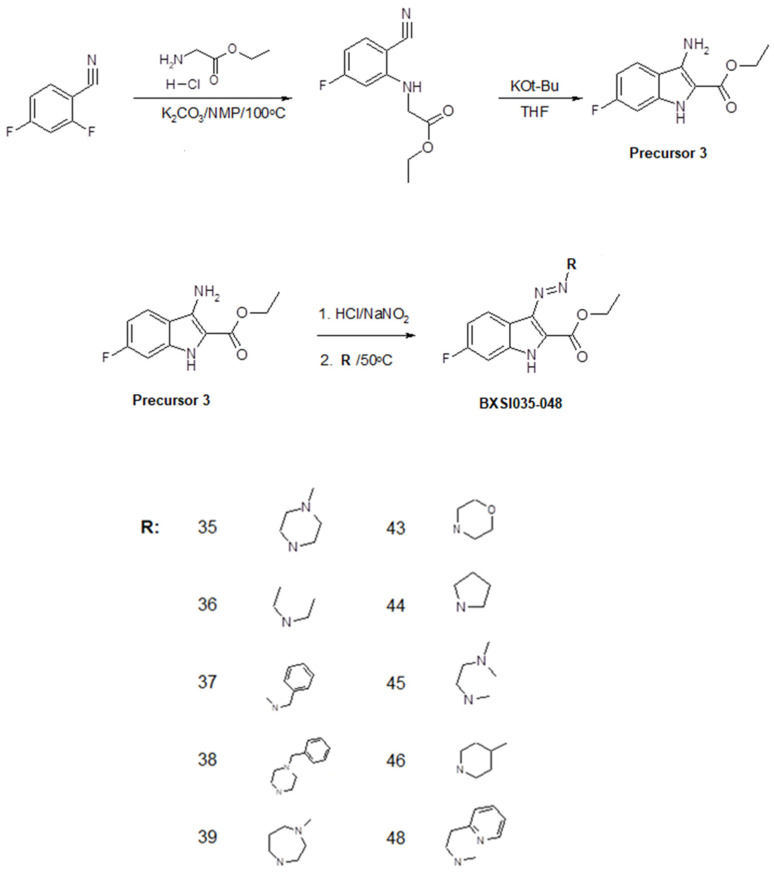
Synthesis of compounds BX-SI035, 036, 037, 038, 039, 040, 043, 044, 045, and 048.

**Figure 5 ijms-26-01870-f005:**
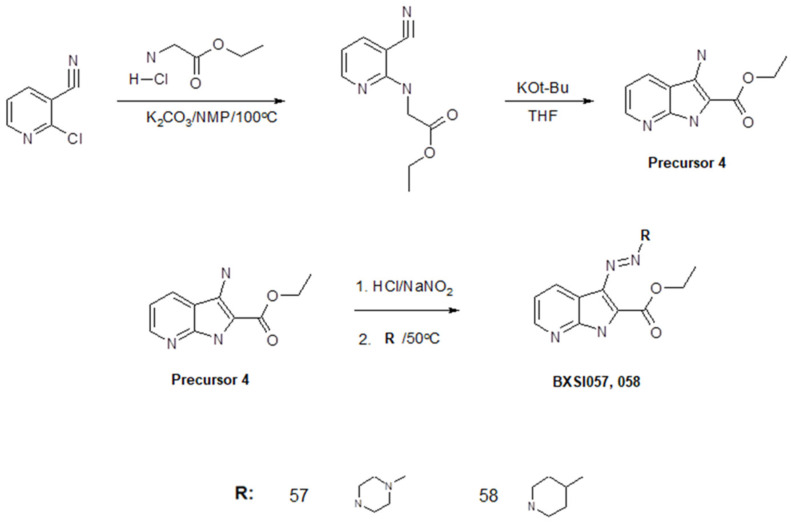
Synthesis of compounds BX-SI057 and 058.

**Figure 6 ijms-26-01870-f006:**
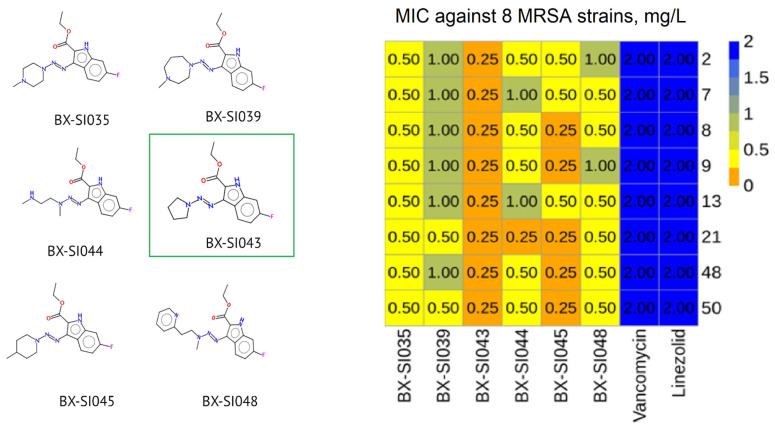
Structures and antimicrobial activity of the most potent compounds. Leading candidate BX-SI043 is highlighted in green.

**Figure 7 ijms-26-01870-f007:**
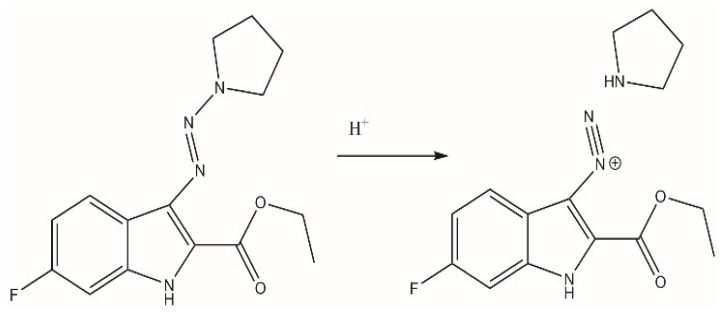
Decomposition of BX-SI043 in acidic conditions.

**Table 1 ijms-26-01870-t001:** Antimicrobial activity and cell toxicity of tested compounds.

Compound	MIC Range Against 8 MRSA Strains, mg/L	IC50 HEF, mg/L
BX-SI001	1	19
BX-SI003	1	3
BX-SI005	1	12
BX-SI010	1	19
BX-SI016	>1	34
BX-SI019	>1	ND
BX-SI020	>1	ND
BX-SI021	>1	ND
BX-SI027	>1	ND
BX-SI035	0.5	20
BX-SI036	>1	ND
BX-SI037	1	ND
BX-SI038	>1	ND
BX-SI039	0.5–1	19
BX-SI040	1	12
BX-SI043	0.25	19
BX-SI044	0.25–1	19
BX-SI045	0.25–0.5	29
BX-SI048	0.5–1	13
BX-SI055	>1	16
BX-SI057	0.5–1	3
BX-SI058	1	12

ND represents a compound that was non-toxic enough in tested concentrations to enable the proper determination of IC50.

**Table 2 ijms-26-01870-t002:** Cytotoxicity and selectivity index for the original compound and most potent derivatives.

Compound	IC50, mg/L		Selectivity Index
	HEF	HepG2	
BX-SI001	19	22	19
BX-SI035	20	21	41
BX-SI039	19	16	20
BX-SI043	19	19	76
BX-SI044	19	13	27
BX-SI045	29	22	74
BX-SI048	13	22	28

**Table 3 ijms-26-01870-t003:** Effects of intragastric administration of various doses of BX-SI043 on rats.

Sex	Dose, mg/kg	Deaths (Dead/ Total)	Changes						
			Behavior	Appearance	Feed Intake	Water Intake	Body Weight	Organ Weight	Organ Histology
Male	0	0/4	no	no	no	no	no	no	no
	300	0/6	no	no	no				no
	600	0/6	no	no	no				yes
	1000	1/6	yes	no	yes				yes
	2000	3/6	yes	yes	yes				yes
Female	0	0/4	no	no	no	no	no	no	no
	300	0/6	no	no	no		no	no	no
	600	0/6	no	no	no		no	no	yes
	1000	1/6	yes	no	yes		no	no	yes
	2000	1/6	yes	yes	yes		no	no	yes

## Data Availability

All data supporting the findings of this study are available within the paper and its Appendix A.

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
