# Peer review of "Novel Triazeneindole Antibiotics: Synthesis and Hit-to-Lead Optimization"

_ijms, 2025, doi:10.3390/ijms26051870_

Round 1
Reviewer 1 Report
Comments and Suggestions for Authors
This paper offers insightful information about the creation of new antibiotics that use triazeneindole derivatives to treat MRSA infections. Even though the research is excellent at its core, focusing on certain aspects will make it much clearer and more comprehensive, which will increase the manuscript's effect.
To support the results, a more thorough statistical analysis of the in vivo investigations is essential. Furthermore, a greater comprehension of these compounds' effectiveness would be possible by extending the conversations on structure-activity correlations. The addition of error bars and statistical significance markers would improve the informative value of a number of figures. Rearranging the supplemental content will also make them easier for readers to access.
The literature review sufficiently discusses current treatments, and the introduction skillfully emphasizes the clinical significance of MRSA infections. However, adding more current references would make the work even more pertinent. The justification for the creation of triazeneindole derivatives is presented in an understandable and compelling manner.
Incorporate thorough statistical analysis techniques for animal research in order to verify the findings.
For better clarity, include the proper error bars with all quantitative values.
To improve the analysis, broaden the conversations about the connections between structure and activity.
Rearrange the supplemental resources to improve accessibility and navigation.
To make sure the literature review is thorough, including more recent sources.
In order to guide future study, offer more in-depth mechanistic insights into the actions that were observed.
To highlight the findings' practical significance, elaborate on possible clinical applications.
To encourage more research in this area, make the section on future directions stronger.
To improve visual clarity, chemical structure drawings should have higher resolution.
To give a clearer context, make sure figure legends are more detailed.
For uniformity and professionalism, format references consistently across the document.
In conclusion, the work makes a substantial contribution to the development of antimicrobial drugs. It will be ready for publishing after taking these suggestions into consideration, opening the door for more developments in the battle against MRSA.
Comments on the Quality of English LanguageGood
Author Response
Comments 1: To support the results, a more thorough statistical analysis of the in vivo investigations is essential. The addition of error bars and statistical significance markers would improve the informative value of a number of figures. Rearranging the supplemental content will also make them easier for readers to access. Incorporate thorough statistical analysis techniques for animal research in order to verify the findings. For better clarity, include the proper error bars with all quantitative values. To improve the analysis, broaden the conversations about the connections between structure and activity.
Rearrange the supplemental resources to improve accessibility and navigation.
Response 1: Thank you so much for such a thorough analysis and valuable comments. We added statistically processed data from animal experiments regarding the effects on animal weight and relative organ weight (lines 223-240 and Supplementary tables 8,9).
Comments 2: Furthermore, a greater comprehension of these compounds' effectiveness would be possible by extending the conversations on structure-activity correlations. he literature review sufficiently discusses current treatments, and the introduction skillfully emphasizes the clinical significance of MRSA infections. However, adding more current references would make the work even more pertinent. To make sure the literature review is thorough, including more recent sources.
In order to guide future study, offer more in-depth mechanistic insights into the actions that were observed. To highlight the findings' practical significance, elaborate on possible clinical applications. To encourage more research in this area, make the section on future directions stronger.
Response 2: We are grateful for these suggestions. We have added more current references (lines 32, 37-43,45) and extended the introduction (lines 57-65). We have extended the further directions section (lines 307-313).
Minor remarks
To improve visual clarity, chemical structure drawings should have higher resolution.
To give a clearer context, make sure figure legends are more detailed.
For uniformity and professionalism, format references consistently across the document.
Response to minor remarks:
Resolution of Figure 1 was improved. References were formatted using MDPI style in Zotero reference manager all across the document.
Reviewer 2 Report
Comments and Suggestions for Authors
The authors presented here the synthesis of a library of 22 triazeneindole derivatives with high activity against a wide panel of multidrug-resistant MRSA clinical isolates. The best in panel compound BX-SI043 showed high activity against 41 multidrug-resistant MRSA strain, relatively low toxicity, providing a great potent antibiotic candidate for MRSA treatment. However, in order to unambiguously determine the actual structure of compounds prepared, the authors need to provide full characterizations of synthesized compounds, including 1H NMR, 13C NMR, HRMS, instead of simply providing data of 1H NMR.
Other minor issues:
The structure of BX-SI037, 040, 043, and 048 need to be revised in Figure 1, F is rotated
The language needs to be carefully checked, like in line 126 'where it shown 4-8 times lower'.
Comments on the Quality of English LanguageNeed to be improved
Author Response
Comments 1: (1) However, in order to unambiguously determine the actual structure of compounds prepared, the authors need to provide full characterizations of synthesized compounds, including 1H NMR, 13C NMR, HRMS, instead of simply providing data of 1H NMR.
Response 1: We agree with this comment.
Unfortunately, we didn't have enough time to do 13C NMR spectra for all 22 substances due to the New Year holidays. We will describe the 13C NMR spectra in the coming week. We will only be able to provide all spectra in the next version of the revision.
Other minor issues:
The structure of BX-SI037, 040, 043, and 048 need to be revised in Figure 1, F is rotated
The language needs to be carefully checked, like in line 126 'where it shown 4-8 times lower'.
Response to minor remarks:
Figure 1 was revised
The text has been edited.
Compound BX-SI043 demonstrated the highest selectivity index (76) and the most potent activity (0.25 mg/L against all tested strains). Consequently, its antimicrobial activity was further evaluated on a broader panel of 41 multidrug-resistant MRSA clinical isolates (Supplementary Table 3), where it exhibited MIC values that were 4-8 times lower compared to the original molecule (Supplementary Table 4).
Reviewer 3 Report
Comments and Suggestions for Authors
Authors described some aspects of the synthesis of triazeneindole analogs and their bioactive applications.
At this stage, however, the manuscript can be considered only a rough draft that requires significant improvement.
(1)
The introduction should be substantially extended, including the application potential of described group of organic compounds, the question of the selection of the strategy for the preparation of target compounds, as we as, general reason of the research in the field of the properties of this compounds class.
(2)
Description of 1H NMR spectrums: all signsls should be precisely described and connected with respective atoms in molecules
Next 13C NMR spectrums should ba also added and, next, analogously described.
(3)
Yields and mp's for synthetised compounds shouls be collected in the table. Next, measuring mp's should be compared with respective data in the literature.
Other minor remarks:
(a) Fig. 2-5.
The carbon atom in the nitrile group exhibit sp hybridisation. In the consequence the C-CN moiety ehibit linear not bent nature.
(b)
Reaction conditions on all schemes should be presented using unified style: temperature, solvent, time, yield below the arrow.
In the conclusion, i recommend major revision of this paper or reject with the possibility of the resubmission after fundamental improvement.
Author Response
Comments 1: (1)
The introduction should be substantially extended, including the application potential of described group of organic compounds, the question of the selection of the strategy for the preparation of target compounds, as we as, general reason of the research in the field of the properties of this compounds class.
Response 1: Thank you for pointing this out. The introduction was extended (lines 57-65) and more recent references were added.
Comments 2. (2)
Description of 1H NMR spectrums: all signsls should be precisely described and connected with respective atoms in molecules
Next 13C NMR spectrums should ba also added and, next, analogously described.
Response 2: We agree with this comment
Unfortunately, we didn't have enough time to do 13C NMR spectra for all 22 substances due to the New Year holidays. We will describe the 13C NMR spectra in the coming week. We will only be able to provide all spectra in the next round of the revision.
Comments 3. (3)
Yields and mp's for synthetised compounds shouls be collected in the table. Next, measuring mp's should be compared with respective data in the literature.
Response 3: The yields of the synthesized substances were collected in Supplementary table 2.
Round 2
Reviewer 2 Report
Comments and Suggestions for Authors
As I stated in last round of comment, full characterizations of synthesized compounds, including 1H NMR, 13C NMR, HRMS are required to confirm the structure as the authors described, thus this is not suitable for publication in this form without the essential data.
Author Response
Comment: As I stated in last round of comment, full characterizations of synthesized compounds, including 1H NMR, 13C NMR, HRMS are required to confirm the structure as the authors described, thus this is not suitable for publication in this form without the essential data.
Response: Thank you for pointing this out, your comments have helped us a lot to improve the quality of our study. We added 13C NMR spectra for the synthesized substances (p. 13, paragraph 4.1.2). We also added LRMS-ESI data and melting points for the substances. The methods are described on page 13, lines 386-392.
Reviewer 3 Report
Comments and Suggestions for Authors
The manuscript can be accepted after necessary icluding of mentioned 13C NMR analysis. I would like to suggest the Editors to give the Authors enough time to perform the analysis and enter the results.
Author Response
Comment: The manuscript can be accepted after necessary icluding of mentioned 13C NMR analysis. I would like to suggest the Editors to give the Authors enough time to perform the analysis and enter the results.
Response: Thank you, now we had enough time to obtain the spectra. We added 13C NMR spectra for the synthesized compounds (p. 13, paragraph 4.1.2). We also added LRMS-ESI data and melting points for the substances. The methods are described on page 13, lines 386-392.
Round 3
Reviewer 2 Report
Comments and Suggestions for Authors The authors have added the data for characterization of compounds made, please also provide the processed spectrum file/picture, including 1H and 13C NMR in supporting information for the readers to check, simply readout data in text is not enough. And also, please include the solvents used for recrystallization for melting point characterization, the solvents used can largely change the number obtained. LRMS data are not considered accurate enough to prove the structure as demonstrated, please provide HRMS data instead. After providing sufficient and high-quality characterization data, this paper should be suitable for publication.Author Response
Comments: The authors have added the data for characterization of compounds made, please also provide the processed spectrum file/picture, including 1H and 13C NMR in supporting information for the readers to check, simply readout data in text is not enough. And also, please include the solvents used for recrystallization for melting point characterization, the solvents used can largely change the number obtained. LRMS data are not considered accurate enough to prove the structure as demonstrated, please provide HRMS data instead. After providing sufficient and high-quality characterization data, this paper should be suitable for publication.
Answer: Thank you for you comments. We have added images of all NMR spectra to Supplementary file 1. In addition, we performed HR-MS analysis. The results are reported in sections 4.1.2, 4.1.3, 4.1.4., 4.1.5. for each substance. The HR-MS methodology is described in section 4.1.2, lines 390-395. We have also specified the solvent for determining the melting point for each substance (sections 4.1.2, 4.1.3, 4.1.4., 4.1.5).
Round 4
Reviewer 2 Report
Comments and Suggestions for Authors
The supplementary file containing NMR spectrum is broken and can't be open, please re-upload the file
Author Response
Comment: The supplementary file containing NMR spectrum is broken and can't be open, please re-upload the file
Response: The supplementary file was not broken but it was wrong. I have uploaded a new Supplementary file 1 that first shows the 1H NMR spectrum then the 13C spectrum for each compound.
Round 5
Reviewer 2 Report
Comments and Suggestions for Authors
The NMR spectrum looks good for 1H NMR, while it's missing the 13C NMR spectrum which was also included in the manuscript, which is also essential for the characterization of compounds. The authors need to include the clean 13C NMR spectrum in their supporting information for the publication.
Author Response
Comment: The NMR spectrum looks good for 1H NMR, while it's missing the 13C NMR spectrum which was also included in the manuscript, which is also essential for the characterization of compounds. The authors need to include the clean 13C NMR spectrum in their supporting information for the publication.
Responce: I apologize for the wrong pdf file. I have uploaded a new Supplementary file 1 that first shows the 1H NMR spectrum then the 13C spectrum for each compound.
Round 6
Reviewer 2 Report
Comments and Suggestions for Authors
Everything looks good and is suitable for publication